# FluxnetLSM R package (v1.0): A community tool for processing FLUXNET data for use in land surface modelling

5  Anna M. Ukkola[1,3], Ned Haughton[1], Martin G. De Kauwe[2], Gab Abramowitz[1,3], Andy J. Pitman[1,3]

[1]ARC Centre of Excellence for Climate System Science, University of New South Wales, Sydney, NSW 2052, Australia

10  [2]Macquarie University, Sydney, NSW 2109, Australia

[3]Climate Change Research Centre, University of New South Wales, Sydney, NSW 2052, Australia

*Correspondence to:* A. M. Ukkola (a.ukkola@unsw.edu.au)

**Abstract.** Flux towers measure ecosystem-scale surface-atmosphere exchanges of energy, carbon dioxide and water vapour. The network of flux towers now encompasses ~900 sites, spread across every continent. Consequently, these data have become an essential benchmarking tool for land surface models (LSMs). However, these data as released are not immediately usable for driving, evaluating and 20  benchmarking LSMs. Flux tower data must first be transformed into a LSM-readable file format, a process which involves changing units, screening missing data and varying degrees of additional gap-filling. All of this often leads to an under-utilisation of these data in model benchmarking. To resolve some of these issues, and to help make flux tower measurements more widely used, we present a reproducible, open-source R package that transforms the FLUXNET2015 and La Thuile data releases 25  into community standard NetCDF files that are directly usable by LSMs. We note these data would also be useful for any other user or community seeking to independently quality control, gap fill or use the FLUXNET data.

30  **1 Introduction**

Land surface models (LSMs) provide the lower boundary condition for climate and weather forecast models, simulating the exchange of carbon, water and energy fluxes between the soil, vegetation and the atmosphere (Pitman, 2003). Flux towers measure ecosystem-scale exchanges of carbon dioxide, 35  water vapour fluxes and energy (Baldocchi, 2014) and have proven invaluable for LSM evaluation and benchmarking (Abramowitz et al., 2008; Best et al., 2015; Blyth et al., 2010; Haughton et al., 2016; Luo et al., 2012; Williams et al., 2009). Flux towers are particularly useful for modelling applications as they provide simultaneous observations of the meteorological data needed for forcing offline models

as well as the key ecosystem variables against which models may be evaluated (e.g. sensible and latent heat) at time intervals similar to those used by LSMs, often over multiple years. As such, they are ideal for characterising the interactions between climate and ecosystem processes and allow the evaluation of LSMs over time periods ranging from sub-daily through to seasonal and inter-annual time scales (e.g.

Blyth et al., 2010; Bonan et al., 2011; Mahecha et al., 2010; Matheny et al., 2014; Powell et al., 2013; Ukkola et al., 2016; Wang et al., 2011; Whitley et al., 2016). The investment in flux tower measurements is considerable and there are multiple benefits to these data being more widely used. First, the use of these data for LSM evaluation and benchmarking helps realise the value of existing investments. Second, where flux tower measurements identify biases in how LSMs represent processes,

the potential exists to improve how well these models simulate the surface energy, water and carbon balances. Since LSMs are central to the simulation of key phenomena including droughts, water resource availability, carbon storage and feedbacks on heatwaves this has direct policy implications. Thirdly, greater use of flux tower measurements by the LSM and climate science community could help with the argument in support of on-going resourcing of flux tower measurements. In short, the

effective and widespread use of flux tower measurements is beneficial across the science and policy communities.

Before data from flux tower sites can be used in models they commonly require significant pre-processing. In principle, flux towers provide near-continuous observations of ecosystem fluxes but, in

practice, the measurements often include discontinuities due to instrument failure or unfavourable weather conditions (Reichstein et al., 2005). As LSMs must be provided with continuous meteorological forcing data, flux tower datasets require varying degrees of gap-filling of missing time steps. This also poses challenges for using these data for model evaluation and benchmarking. Ideally, models should be evaluated against high-quality observations. Due to data gaps, as well as

measurement biases (e.g. Leuning et al., 2012), flux tower measurements do not provide reliable observations representative of the true ecosystem dynamics in all circumstances. Arguably therefore, the full breadth of flux tower data available across the entire network is unlikely to be suitable to the role of evaluating LSMs.

FLUXNET, an international network of flux tower sites, comprises of >900 sites globally (http://fluxnet.fluxdata.org/). The latest FLUXNET data release (FLUXNET2015; http://fluxnet.fluxdata.org/data/fluxnet2015-dataset/) provides flux tower measurements for 212 sites. It was preceded by the La Thuile Synthesis Dataset (http://fluxnet.fluxdata.org/data/la-thuile-dataset/), which comprises of 252 flux tower sites, 141 of which are not currently available in FLUXNET2015.

The available data overcome some of the limitations of raw eddy covariance measurements through significant post-processing and gap-filling. Despite this, these datasets cannot be employed directly by LSMs. Critically, not all FLUXNET data releases are provided with temporally continuous observations of all essential meteorological variables (e.g. precipitation and wind speed) for forcing LSMs. For example, across 155 FLUXNET2015 "FULLSET" open data policy (Tier 1) sites reporting

half-hourly observations, nearly all sites include gaps in rainfall and 77% of the sites have missing air

temperature observations with up to 61% (median 5%) of the time series missing despite this variable being nominally gap-filled. Further, evaluation variables, such as latent and sensible heat, are generally gap-filled but to vastly different extents depending on the site and variable. For example, between 0% and 89% (median 31%) of the latent heat time series and 0% and 83% (median 25%) of the sensible heat time series have been gap-filled across the 155 sites. This poses a challenge for utilising these data for LSM applications and additional post-processing is necessary. A specific concern is that individual land surface modellers are very likely to post-process flux data in different ways, with different assumptions and varying levels of acceptance on how many gaps represent a worthwhile data set. When the gap-filled data are subsequently used and published, the detail of how all the possibilities around post-processing the data are resolved is rarely fully documented. This leads to difficulties in interpreting model evaluation studies, a lack of reproducibility and, given many groups process data individually, wasted effort.

In an effort to resolve some of these problems and to connect the flux tower researchers with the LSM researchers more strongly, we present the R package "FluxnetLSM" to facilitate the processing of FLUXNET datasets for use in LSMs. The package serves several important functions. Firstly, it enables the creation of fully gap-filled meteorological forcing datasets for running LSMs. Past studies have relied on various (often ad-hoc) gap-filling methods that are rarely fully documented in the literature. Worryingly, it would be virtually impossible to reproduce many existing LSM evaluation and benchmarking studies although we note some exceptions (Best et al., 2015). The R package provides a community tool for creating LSM forcing datasets in a fully citeable and reproducible framework. Secondly, the package assists with the quality controlling of the data. It enables the selection of good-quality measurement periods and sites through automated screening of heavily gap-filled or missing data periods according to user-defined thresholds. To complement the automated quality controlling, the package also provides tools for creating diagnostic plots to visualise output data periods. This facilitates the detection of data periods with unusual variability or variables exhibiting unusual magnitudes. Finally, the package converts the flux tower data into the community standard NetCDF format used by the climate modelling and LSM community and collates metadata on data variables, flux tower sites as well as processing steps in the output files.

The package offers a useful tool for post-processing eddy covariance datasets for modelling applications and simplifies rigorous documentation of data processing methods in LSM studies to enhance their reproducibility. Specifically, future studies using these data would be able to explicitly demonstrate how the data were used, gap-filled, quality controlled and so on, and this could be reproduced by other users. In the following sections, we describe the different functionalities of the package.

**2 Package description**

The `FluxnetLSM` package (v1.0) was developed to serve as a community tool to facilitate the use of flux tower measurements in LSMs. It is written in the open-source R language (https://www.r-project.org/) and is freely accessible in a version-controlled repository (see Code Availability for full details). Instructions for installation are provided in the following section.

The package has two processing streams: the collection of site metadata and processing of high frequency temporally varying variables. These are described in sections 2.3 and 2.4, respectively. The package outputs a separate NetCDF file for meteorological and evaluation variables, with metadata stored in each file. Additionally, a log file is produced detailing output file names, potential warnings and errors. The package also provides the option to produce diagnostic plots for further data exploration. Figure 1 illustrates the general workflow with each component described in detail below.

## 2.1 Installation and requirements

`FluxnetLSM` requires R version ≥3.1.0. It relies on base R functions as well as three additional packages: `R.utils`, `ncdf4` and `rvest`. These packages should be installed prior to the installation of `FluxnetLSM`. The `devtools` package is also recommended to aid installation.

The `R.utils`, `ncdf4`, `rvest` and `devtools` packages can be installed directly in R with the

command `install.packages("package_name")`. The `FluxnetLSM` package can be downloaded from the Github repository at https://github.com/aukkola/FluxnetLSM and installed within R by typing:

```
devtools::install_github("aukkola/FluxnetLSM")
```

Alternative installation methods are provided in the package github repository. After installation, the `FluxnetLSM` package can be loaded into the R session by typing `library(FluxnetLSM)`. Other required packages are loaded automatically by the `FluxnetLSM` package.

## 2.2 Running FluxnetLSM

The package is run by invoking a single R function called `convert_fluxnet_to_netcdf`:

```
convert_fluxnet_to_netcdf(site_code, infile, era_file=NA, out_path,
                          conv_opts=get_default_conversion_options(),
                          plot=c("annual", "diurnal", "timeseries"),
                          ...)
```

The user must set three arguments (`infile`, `site_code` and `out_path`), with all other arguments

being optional. Each argument and its default value is described in Table 1 and discussed in detail in

the following sections. A full example for usage is provided in Section 3. Three example scripts are also provided with the package and are stored in `examples/FLUXNET2015` and `examples/LaThuile` for the FLUXNET2015 and La Thuile data releases, respectively. In each directory, the `example_conversion_single_site.R` file shows an example for processing a

5  single site. The `example_conversion_multiple_sites.R` and `example_conversion_multiple_sites_parallel.R` files show an example for processing multiple sites using serial and parallel programming, respectively.

**2.3 Collation of site metadata**

The package collates metadata on the flux tower sites and stores these as attributes in the output NetCDF files. These include information required for modelling such as site coordinates, elevation and vegetation type. The primary source for metadata is a site attribute file provided with the package (stored in `data/Site_metadata.csv`). This file includes metadata detailed in Table 2 for the Tier

1 sites of the FLUXNET2015 November 2016 release (see http://fluxnet.fluxdata.org/data/fluxnet2015-dataset/ for more information). The metadata were collated by the code developers from the site information provided on the FLUXNET website as well as individual flux tower network websites (see `data/README.md` for full details). The metadata file can be edited by the user to include additional sites or to modify existing data. The code first extracts site metadata from the CSV file. If any metadata

cannot be found in the provided file, the code attempts to retrieve missing metadata from the FLUXNET website (http://fluxnet.fluxdata.org), followed by the Oak Ridge National Laboratory (ORNL) FLUXNET website (https://fluxnet.ornl.gov/) by using functions for reading html webpages provided in the `rvest` library.

Additionally, the code stores the dataset name and version (as set by the `datasetname` and `datasetversion` arguments to the main function), as well as the processing options, time and date as attributes in the output files. The code also calculates the mean annual precipitation for the output period when precipitation is outputted. It is stored as an attribute in the meteorological output file and can be useful particularly for rescaling precipitation for LSM spin-up so that each year's precipitation

during the spin-up matches the site average.

This processing step connects key site metadata directly to each model forcing files. It can be extended to include additional metadata, such as site soil or vegetation properties, with minimal code modifications. For example, LSMs generally use plant functional types (PFT) instead of the

International Geosphere-Biosphere Programme (IGBP; http://www.igbp.net/) vegetation types automatically retrieved by the package (Poulter et al., 2011). An example is provided for writing the PFT type for the CABLE LSM (Wang et al., 2011) and can be invoked by setting the `model` argument to the desired model name. Full instructions for adding model-specific parameters are provided in the package README file.

### 2.4 Processing of high frequency data variables

#### 2.4.1 Output variables

The package is supplied with a suggested list of output variables that will be processed by the package for each site, where available. Separate lists are provided for FLUXNET2015 FULLSET and SUBSET, and La Thuile data releases due to different naming conventions and variables (stored in `data/Output_variables_FLUXNET2015_FULLSET.csv`, `data/Output_variables_FLUXNET2015_SUBSET.csv` and `data/Output_variables_LaThuile.csv`, respectively) The output variables are categorised as meteorological or evaluation variables, and a separate NetCDF output file is produced for each category. Where possible, the output variables are named using the Assistance for Land-surface Modelling Activities (ALMA) convention (http://www.lmd.jussieu.fr/~polcher/ALMA/convention_output_3.html) commonly employed by LSMs. The package also performs common unit conversions between the original FLUXNET and ALMA convention units (see section 4.4). The output variables are fully customisable according to user requirements by removing or adding variables to the output variable list. The information required for each output variable is shown in Table 3.

#### 2.4.1.1 Meteorological variables

The meteorological variables include the data variables typically required to force LSMs. The meteorological variables processed by the package by default are detailed in Supplementary Table 1. The user can also nominate essential meteorological variables that must be available and processed by modifying the `Essential_met` field in the output variable list (see Table 3). By default, these include air temperature, downward shortwave radiation (or photosynthetically active radiation), vapour pressure deficit, precipitation and wind speed. If any of these variables are not provided in the input data file, the code will terminate and the site will not be processed. The code provides several options for gap-filling meteorological variables if required (see Section 2.4.3 for details).

#### 2.4.1.2 Evaluation variables

The evaluation variables include the data variables typically predicted by land surface models and used to evaluate model outputs. The default evaluation variables processed by the package are provided in Supplementary Table 2. The user can nominate preferred evaluation variables by modifying the `Preferred_eval` field in the output variable list (see Table 3). By default these include net radiation, latent (LE) and sensible (H) heat and net ecosystem exchange (NEE). If none of the preferred variables are available in the input data file, the site will not be processed. The evaluation variables can be gap-filled by the package using statistical methods (Section 2.4.3).

In addition to common evaluation variables, the package also processes and outputs uncertainty estimates provided with the FLUXNET2015 release by default. These include uncertainty bounds for LE, H and NEE, as well as error estimates for gross primary productivity (GPP). Several estimates for NEE and GPP are also included to reflect the inherent uncertainties in deriving these variables from eddy covariance data (Papale et al., 2006; Reichstein et al., 2005; Supplementary Table 2).

### 2.4.2 Gap-filled and missing values

The code produces NetCDF files with whole years of data only, to ensure LSM automated spin-up procedures remain relatively unbiased. It determines which years are included in its output according to user-defined thresholds for gap-filled and missing values as detailed below.

A threshold must be set for the maximum percentage of missing values per year (argument `missing`, 15% by default). The code checks for the percentage of missing values for each data variable during each year. If *any* essential meteorological variables or *all* preferred evaluation variables have missing values in excess of this threshold, the year is not processed.

Additionally, thresholds can be set for the maximum percentage of all gap-filling (default option; set by argument `gapfill_all` using 20% as the default) or separately for "good", "medium" and "poor" quality gap-filling (arguments `gapfill_good`, `gapfill_med` and `gapfill_poor`, respectively; see section 4.3). The percentage of gap-filled values is then checked for each data variable with a corresponding quality control flag during each year. If *any* essential meteorological variable or *all* preferred evaluation variables include gap-filled values in excess of the threshold(s), the year is not processed. Note the November 2016 FLUXNET2015 release has gaps in quality control flags for latent and sensible heat variables even when data are present. A fix has been provided (http://fluxnet.fluxdata.org/data/fluxnet2015-dataset/known-issues/) but if not implemented, the data quality cannot be ascertained from the flags (D. Papale, pers. comm.) and is treated by the package as poor-quality gap-filling.

If a threshold for gap-filling is set, the percentage of both gap-filled and missing values must not exceed their respective thresholds for a year to be processed. If no years fulfilling the criteria are found, or the time period is shorter than the user-defined minimum number of consecutive years (set by argument `min_yrs`, by default 2 years), the site it not processed. If several, non-consecutive, time periods fulfilling the criteria are found, these are written to separate output files.

Provided that at least one evaluation variable has fewer gaps than the user-defined thresholds, all evaluation variables are written to the output file by default, with the exception of any variables that only contain missing values. An option is provided to discard any evaluation variables with gaps exceeding the user-defined thresholds by setting the argument `include_all_eval` to FALSE.

### 2.4.3 Gap-filling variables

LSMs require continuous forcing data, but a number of essential meteorological variables (rainfall, wind speed, incoming longwave radiation and air pressure) are not fully gap-filled in the FLUXNET2015 "FULLSET" and/or La Thuile releases. The package provides two methods for gap-filling meteorological variables: statistical and ERA-Interim (Dee et al., 2011; Vuichard and Papale, 2015). Additionally, statistical methods are provided for gap-filling evaluation variables.

### 2.4.3.1. ERA-Interim –based gap-filling

Downscaled ERA-Interim reanalysis estimates are provided as part of the FLUXNET2015 dataset for gap-filling meteorological variables. These are available only in the "FULLSET" version of the FLUXNET2015 release (http://fluxnet.fluxdata.org/data/fluxnet2015-dataset/fullset-data-product/), whereas the "SUBSET" version of the dataset has already been gap-filled using ERA-Interim but offers the user less flexibility for controlling for gap-filling quality (with missing, medium- and poor-quality gapfilled time steps readily gapfilled with ERA-Interim).

This gapfilling option is chosen by setting the argument `met_gapfill` to "ERAinterim" and by providing the name of the ERAinterim input file to argument `era_file`. The ERA-Interim variable corresponding to each meteorological variable is set in the output variable list (`ERAinterim_variable` field; Table 2). If an ERA-Interim estimate is available for a given variable, the code gap-fills any missing time steps with the corresponding ERA-Interim data value. The package saves information on the gap-filled time steps in quality control flag variables (see Section 2.4.4 for details).

### 2.4.3.2 Statistical gap-filling

Alternatively, meteorological, as well as evaluation, variables can be gap-filled using statistical methods using a combination of methods depending on the length of missing periods. This gap-filling option can be chosen for meteorological and evaluation variables by setting arguments `met_gapfill` and `flux_gapfill` to "statistical", respectively.

Surface air pressure and incoming longwave radiation are synthesised using empirical functions (Abramowitz et al., 2012). Air pressure is calculated from air temperature and elevation using the barometric formula as detailed in Supplementary Section S.1.1. Three methods for synthesising longwave radiation are provided ("Abramowitz_2012", "Swinbank_1963" and "Brutsaert_1975") and are set by the argument `lwdown_method`. "Swinbank_1962" calculates longwave radiation based on air temperature, whereas "Abramowitz_2012" (default) and "Brutsaert_1975" calculate it from air temperature and relative humidity. Each of these methods is detailed in Supplementary Section S.1.2.

For all other meteorological and evaluation variables, short data gaps (by default up to 4 hours, set by argument `linfill`) are first gap-filled using linear interpolation between the previous and next available time steps. This prevents the introduction of abrupt variations, but leads to a loss of some subdiurnal variability.

For meteorological variables, longer gaps (by default up to 10 days, set by argument `copyfill`) are then gap-filled by taking the average of the corresponding time steps during other years (Blyth et al., 2010). Data gaps that are longer than set by `copyfill` are not gap-filled due to the limitations of statistical gap-filling for stochastic variables, such as rainfall.

For evaluation variables, longer gaps (by default up to 30 days, set by argument `regfill`) are gap-filled using a linear regression of each evaluation variable against one or several meteorological variables (adapted from Best et al., 2015). When incoming shortwave radiation, air temperature and humidity (relative humidity or vapour pressure deficit) are available, the code will perform a multiple linear regression against these variables. Else, if only shortwave radiation is available, a linear regression against this variable is performed. All available time steps are used to construct a linear regression model separately for day- and night-time (using incoming solar radiation of 5 W m$^{-2}$ as the day-night threshold; Abramowitz et al., 2012). The linear regression models are then used to predict missing values at each time step. If none of the meteorological variables are available, or data gaps are longer than set by `regfill`, the evaluation variables are not gap-filled. If `copyfill` is preferred over `regfill`, the code will default to this option if `regfill` is set to `NA`.

After performing the gap-filling, the code checks for missing values (as per Section 2.4.2). If missing values remain in *any* essential meteorological variables or *all* preferred evaluation variables at a given year, the year is removed from the outputs. If the remaining time period is shorter than the user-defined minimum number of consecutive years, the site is not processed.

**2.4.4 Quality control flags**

30    The code retains and outputs the original FLUXNET quality control (QC) flags, when these are included in the output variable list. These flags are set to 0 for measured data, and 1, 2 and 3 for good, medium and poor quality gap-filling, respectively, for La Thuile and FLUXNET2015 "FULLSET" data (Reichstein et al., 2005; http://fluxnet.fluxdata.org/data/fluxnet2015-dataset/). FLUXNET2015 "SUBSET" QC flags are as per "FULLSET" for measured and good-quality gapfilled data, with flags set to 2 for ERA-Interim gapfilled time steps.

Additionally, the code produces QC flags for meteorological variables when they are gap-filled using ERA-Interim data or statistical methods. The QC flag is set to 4 when a time step is gap-filled with ERA-Interim data and 5 for statistical gap-filling. If a QC flag does not exist for a given variable, the code creates a QC flag variable with measured time steps set to 0 and ERA-Interim or statistically gap-

filled time steps set to 4 or 5, respectively. This flag is automatically stored as a variable in the meteorological data output file and is named as the output variable plus the extension "_qc" (e.g. Precip_qc). See below for QC flag conventions when aggregating data to coarser time steps.

**2.4.5 Aggregation to coarser time steps**

By default, the package outputs the data in its original time resolution. However, a longer time step may be desired for some model applications. The package allows the aggregation of the data to up to a daily resolution. The aggregated time step size (in hours) is set by the argument `aggregate` and can
be any number between the original resolution (usually 30 minutes) and 24 hours (daily), as long as it is divisible by 24 to allow a regular number of time steps to be aggregated. If any of the time steps being aggregated are missing, the new coarser time step will also be set to missing. The QC flags (if outputted) are assigned a fraction between 0-1, indicating the percentage of time steps used for aggregation that were observed.

**2.4.6 Unit conversions**

The package uses ALMA convention units for outputs by default where possible (as indicated in Supplementary Tables 1 and 2). These differ from the original FLUXNET units for a number of
variables and a conversion is performed in each case. Available conversions are detailed in Table 4. If a conversion is not available for the specified units, the code will produce an error and terminate. Additionally, the package provides functions for converting i) vapour pressure deficit to relative humidity, ii) relative humidity to specific humidity and iii) photosynthetically active radiation (PAR) to incoming shortwave radiation ($SW_{down}$).

For these conversions, saturated vapour pressure ($e_{sat}$) is first calculated from air temperature ($T_{air}$; °C) (Jones, 1992) at each time step as

$$e_{sat} = 613.75 * \exp[17.502 * T_{air} / (240.97 + T_{air})] \tag{1}$$

Relative humidity ($R_h$; %) is then determined from $e_{sat}$ and vapour pressure deficit ($D$; Pa) as

$$R_h = 100 * (1 - (D * 100) / e_{sat}) \tag{2}$$

To calculate specific humidity ($Q_{air}$; kg kg$^{-1}$), specific humidity at saturation ($w_s$; kg kg$^{-1}$) is derived from $e_{sat}$ and air pressure ($\rho_{air}$; Pa) as

$$w_s = 0.622 * e_{sat} / (\rho_{air} - e_{sat}) \tag{3}$$

$Q_{air}$ is then calculated as

$$Q_{air} = (R_h/100) * w_s \qquad (4)$$

PAR ($\mu$mol m$^{-2}$ s$^{-1}$) is converted to SWdown (W m$^{-2}$) following Monteith and Unsworth (1990):

$$SW_{down} = PAR * (1/2.3) \qquad (5)$$

Negative PAR values are set to 2.17 W m$^{-2}$ (equivalent to 5 $\mu$mol m$^{-2}$ s$^{-1}$) to avoid problems forcing LSMs with negative SW$_{down}$.

**2.4.7 Visualisation of outputs**

The package provides an option to visualise outputs variables. Three types of plots can be produced: a mean annual cycle, a mean diurnal cycle by season and a time series figure. This is controlled by the argument `plot` that can be set to any combination of `annual`, `diurnal` and `timeseries` for the three plot options, respectively. Examples of each plot are provided in Figure 2.

The outputs are retrieved from the output NetCDF files and all data variables are plotted with separate figures produced for meteorological and evaluation variables. Any missing values are ignored during plotting, but their presence is noted in the figure, when applicable. The data are plotted in their output units, with the exception of air temperature (converted from Kelvin to Celsius) and rainfall (converted from mm/s to mm/time step). It is envisaged the plots will complement the automated quality control performed during data processing and enable further detection of unsuitable data periods or sites.

**3 Example application**

Here we present an example application using "FluxnetLSM" for processing FLUXNET2015 "FULLSET" data at the Howard Springs (Australia) flux tower site. This example is provided in full with the package and stored in `examples/FLUXNET2015/example_conversion_single_site.R`. It is also reproduced in Supplementary section S.2 for convenience. Meteorological data is gap-filled using ERA-Interim estimates in this example but this functionality can be disabled if desired by setting `met_gapfill` argument to `NA` (see below). The user must provide four inputs, with the following inputs used in this example:

```
infile    <- "FLX_AU-How_FLUXNET2015_FULLSET_HH_2001-2014_1-3.csv"
ERA_file  <- "FLX_AU-How_FLUXNET2015_ERAI_HH_1989-2014_1-3.csv"
site_code <- "AU-How"
out_path  <- "~/FluxnetLSM/Outputs"
```

The data can then be processed by invoking:

```
convert_fluxnet_to_netcdf(infile, site_code, ERA_file out_path,
                          met_gapfill="ERAinterim")
```

All other arguments are left to their default values in this example (see Table 1 for argument descriptions). The package automatically selects output years based on the default thresholds (as detailed in Section 2.3.2). Figure 3 shows the full time series of essential meteorological variables and two example evaluation variables at Howard Springs. The code helps exclude time periods with extensive missing periods, such as the first year (2001) of the time series, as well as heavily gap-filled time periods (e.g. around January 2007). Extended periods with missing QC flags (see Section 2.2.3) are also excluded for evaluation variables due to unknown data quality (Figure 3b). Based on the default thresholds, the time period 2010-2014 is chosen and outputted, indicated by grey shading in Figure 3. The rest of the data are discarded. Thresholds can of course be modified by the user to change this result.

Once the data have been processed and outputted, they can be visualised. Three types of plots are produced by default: mean annual and diurnal cycles and a time series plot. Figure 2 shows an example of each type of output plot produced by the package. These plots can be used for further quality controlling to detect any anomalous data periods not automatically excluded by the package.

**4 Discussion and Conclusions**

Efforts to better utilise existing observational data provide multiple benefits, including bringing research communities together, evaluating models against broader data, and providing further support to groups seeking to maintain primary observations. To maximise the use of observed data by communities other than those that collect the data, it is advantageous to make the data as accessible and easy to use as possible. In the case of the FLUXNET data, one major community is the land surface modelling sciences. Land surface models are key components in climate modelling and are therefore critical to broader science and policy communities. It is important to take any opportunities to improve the evaluation of land surface models that exist, and making FLUXNET datasets more reliably and easily available to the land surface modelling community removes a significant hurdle in that process.

To enhance transparency, to help reproducibility and as a platform for further community efforts we have presented an R package that transforms FLUXNET data into a form directly useable by LSMs. As released, FLUXNET data cannot be directly employed in LSMs due to data gaps, incompatible units and non-standard (land surface community) file format (CSV rather than NetCDF). The R package also collates metadata on data processing steps and the flux tower sites and stores these in the output files for easy access, and to permit more reliable reproducibility for modelling experiments. Finally, the

package generates visualisations of outputs to facilitate further quality control of flux tower data and to help inform appropriate site selection, an important step in applying these data to modelling studies.

5 The package is open source, fully documented and simple to use, requiring minimal input from the user. It allows multiple sites to be processed into a form usable by LSMs in a short R script. Simultaneously, it provides optional settings for an advanced user to produce flux tower datasets suited for specific applications. For example, the user may wish to process the data differently if interested in evaluating models during short-term phenomena (such as heat waves) compared to longer seasonal to 10 annual scales. Importantly, the package provides a tool for producing flux tower datasets for modelling applications in a fully citeable and reproducible framework. The package is stored in a publicly available repository and is being actively developed with community contributions encouraged.

**Code availability**

15 The `FluxnetLSM` code can be downloaded from the Github repository at https://github.com/aukkola/FluxnetLSM. Other required packages (`R.utils`, `ncdf4` and `rvest`) can be installed directly in R with the command `install.packages("package_name")`. See section 2.1 for further details on installation.

20 **Acknowledgements**

We acknowledge the support of the Australian Research Council Centre of Excellence for Climate System Science (CE110001028). M.G. De Kauwe was supported by Australian Research Council Linkage grant LP140100232. This work used eddy covariance data acquired and shared by the 25 FLUXNET community, including these networks: AmeriFlux, AfriFlux, AsiaFlux, CarboAfrica, CarboEuropeIP, CarboItaly, CarboMont, ChinaFlux, Fluxnet-Canada, GreenGrass, ICOS, KoFlux, LBA, NECC, OzFlux-TERN, TCOS-Siberia, and USCCC. The ERA-Interim reanalysis data are provided by ECMWF and processed by LSCE. The FLUXNET eddy covariance data processing and harmonization was carried out by the European Fluxes Database Cluster, AmeriFlux Management 30 Project, and Fluxdata project of FLUXNET, with the support of CDIAC and ICOS Ecosystem Thematic Center, and the OzFlux, ChinaFlux and AsiaFlux offices.

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

**Tables**

**Table 1**: Input arguments to the main `convert_fluxnet_to_netcdf` function. Conversion
options can be passed directly into the function or retrieved using

5    `get_default_conversion_options()` (see example in Supplementary section S.1).

| Argument | Description | Default value |
|---|---|---|
| `infile` | FLUXNET2015 or La Thuile file(s) containing data variables | - |
| `site_code` | FLUXNET site ID (see Table 2) | - |
| `out_path` | User-defined output path | - |
| `era_file` | FLUXNET2015 file containing ERA-interim variables | NA |
| `conv_opts` | List of conversion options (by default retrieved automatically) | get_default_conversion_ options() |
| `plot` | Output plots to be produced. Set to NA if not required | c("annual", "diurnal", "timeseries") |
| **Conversion options:** | | |
| `datasetname` | Name of the dataset being processed (FLUXNET2015 or LaThuile) | FLUXNET2015 |
| `datasetversion` | User-defined dataset version. Stored as metadata in output files. | n/a |
| `flx2015_version` | FLUXNET2015 version (FULLSET or SUBSET) | FULLSET |
| `fair_use` | La Thuile data policy/ies the output dataset should comply with. | Fair_Use |
| `fair_use_vec` | A vector of La Thuile data use policies for each year in the data files. Retrieved automatically from `data/LaThuile_site_policy.csv`. | NA |
| `aggregate` | Time step (in hours) to aggregate data to | NA |
| `met_gapfill` | Method to gapfill meteorological data: "ERAinterim", "statistical" or NA (no gapfilling) | NA |
| `flux_gapfill` | Method to gapfill flux data: "statistical" or NA (no gapfilling) | NA |
| `missing` | Max. percentage of time steps allowed to be missing in any given year | 15 |
| `gapfill_all` | Max. percentage of time steps allowed to be gap-filled (any quality) in any given year | 20 |
| `gapfill_good` | Same as above for good-quality gap-filling | NA |
| `gapfill_med` | Same as above for medium-quality gap-filling | NA |
| `gapfill_poor` | Same as above for poor-quality gap-filling | NA |
| `min_yrs` | Min. number of consecutive years to process | 2 |
| `linfill` | Max. consecutive length of time (in hours) to be gap-filled using linear interpolation | 4 |
| `copyfill` | Max. consecutive length of time (in number of days) to be | 10 |

| | | |
|---|---|---|
| | gap-filled using copyfill | |
| regfill | Max. consecutive length of time (in number of days) to be gap-filled using multiple linear regression | 30 |
| lwdown_method | Method to synthesise incoming longwave radiation. One of "Abramowitz_2012", "Swinbank_1963" and "Brutsaert_1975". | Abramowitz_2012 |
| include_all_eval | Should all evaluation values be outputted, regardless of data gaps? If set to FALSE, any evaluation variables with missing or gap-filled values in excess of the thresholds will be discarded | TRUE |
| model | Name of land surface model. Allows additional model parameters to be stored as metadata in output files | NA |

25

**Table 2:** Site metadata provided with the package. All attributes are provided for each Tier 1 site, with the exception of tower and canopy height.

| Attribute | Description |
| --- | --- |
| SiteCode | FLUXNET site ID*, e.g. AU-How |
| Fullname | FLUXNET site name*, e.g. Howard Springs |
| SiteLatitude | Latitude (degrees north) |
| SiteLongitude | Longitude (degrees east) |
| SiteElevation | Elevation (metres) |
| IGBP_vegetation_short | Short IGBP vegetation type, e.g. WSA |
| IGBP_vegetation_long | Long IGBP vegetation type, e.g. Woody Savannas |
| TowerHeight | Height of measurement tower (metres) |
| CanopyHeight | Height of canopy at site (metres) |
| Tier | FLUXNET2015 site tier* |
| Exclude | Should site be excluded? Allows sites with known problems to be excluded *a priori*. Set to TRUE or FALSE. |
| Exclude_reason | Reason why site should be excluded (user-defined) |

*See http://fluxnet.fluxdata.org/sites/site-list-and-pages/

**Table 3:** Attributes required for each output variable (stored separately for FLUXNET2015 and La Thuile data releases in `data/Output_variables_*.R`).

| Field name | Description | Value |
|---|---|---|
| Fluxnet_variable | Original FLUXNET variable name[1] | e.g. TA_F_MDS |
| Fluxnet_unit | Original FLUXNET variable unit[1] | e.g. C |
| Fluxnet_class | Variable data type. Used to define the `colClasses` argument in the R `read.csv` function when reading the input data file. Set to "numeric" if not known. | "numeric" or "integer" |
| Output_variable | Output variable name | User-defined, e.g. Tair |
| Output_unit | Output unit (note section 2.4.6 for unit conversions) | User-defined, e.g. K |
| Longname | Long variable description. Written as a variable attribute in the output file. | User-defined, e.g. Near surface air temperature |
| Standard_name | Climate and Forecast (CF) convention standard name[2]. Written as a variable attribute in the output file. | User-defined, e.g. air_temperature |
| Data_min | Minimum acceptable data value. Used to check data ranges (using output units). | User-defined, e.g. 200 |
| Data_max | Maximum acceptable data value. Used to check data ranges. | User-defined, e.g. 333 |
| Essential_met | Sets variable as essential when set to TRUE (see section 2.4.1.1) | "TRUE" or "FALSE" |
| Preferred_eval | Sets variable as preferred when set to TRUE (see section 2.4.1.2) | "TRUE" or "FALSE" |
| Category | Determines if the variable is written in the meteorological or evaluation NetCDF output file. | "Met" or "Eval" |
| ERAinterim_variable | Name of ERA-interim variable[1] | e.g. "TA_ERA" |
| Aggregate_method | Method used to aggregate the variable (mean or sum). | e.g. mean |

[1]Must match naming conventions on http://fluxnet.fluxdata.org/data/fluxnet2015-dataset/ for FLUXNET2015 and http://fluxnet.fluxdata.org/data/la-thuile-dataset/ for La Thuile.

[2]see http://cfconventions.org/standard-names.html

**Table 4:** Available unit conversions.

| Variable | Variable name | | Original unit | Converted unit |
|---|---|---|---|---|
| | FLUXNET2015* | La Thuile | | |
| Air temperature | TA_F_MDS | Ta_f | C | K |
| Rainfall | P | Precip_f | mm | kg m$^{-2}$ s$^{-1}$ |
| Air pressure | PA | - | kPa | Pa |
| Atmospheric $CO_2$ concentration* | CO2_F_MDS | CO2 | $\mu$mol $CO_2$ mol$^{-1}$ | ppm |

*Note these units are equal and the conversion is included to allow different notations

*FULLSET variable names reported here

25

30

| Variable | Variable name | | Original unit | Converted unit |
|---|---|---|---|---|
| Air temperature | TA_F_MDS | Ta_f | C | K |
| Rainfall | P | Precip_f | mm | kg m$^{-2}$ s$^{-1}$ |

**Figures**

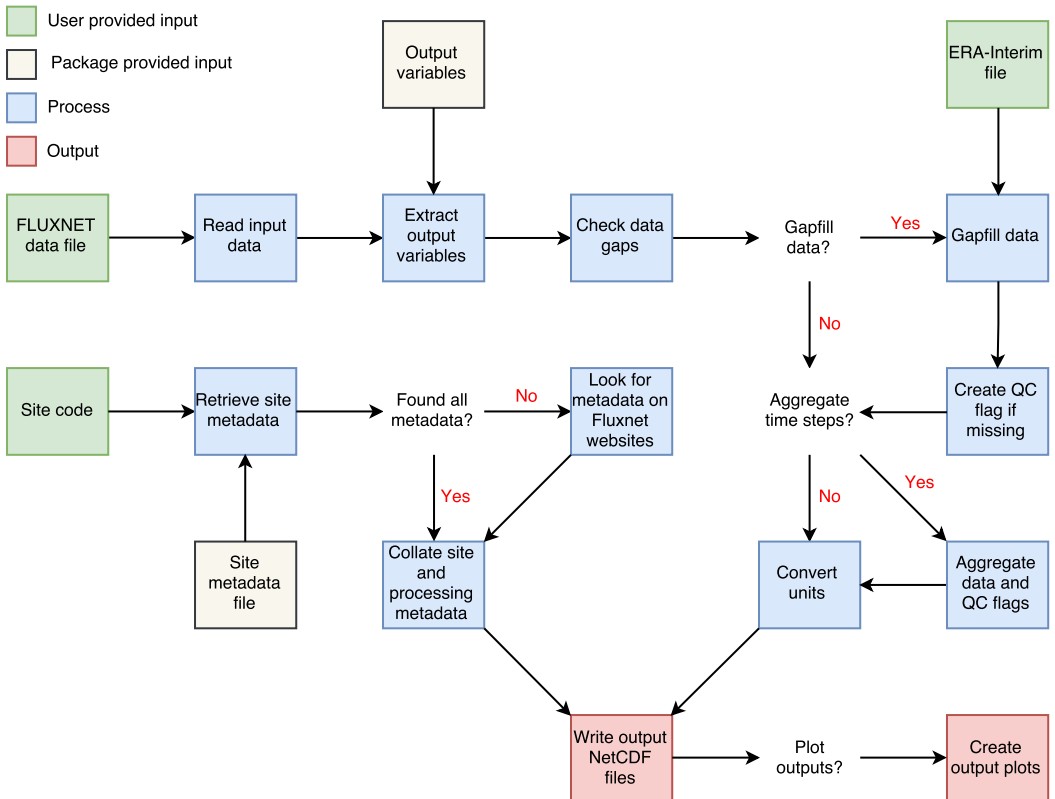

**Figure 1:** General workflow of the FluxnetLSM R package.

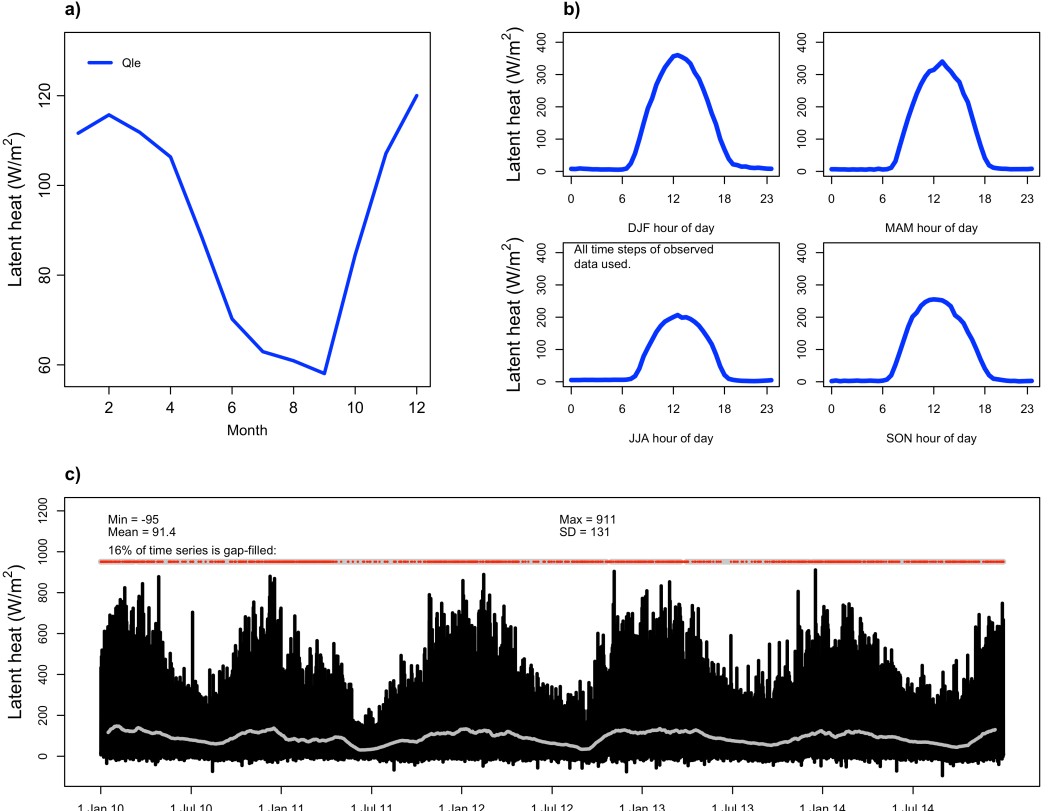

**Figure 2:** Examples of output plots produced by the package. Mean annual cycle by month is shown in panel a) and mean diurnal cycle by season in panel b). A time series is plotted in panel c), with the full time series shown in black and a smoothed 14-day running mean in grey. Gap-filled periods are indicated in red.

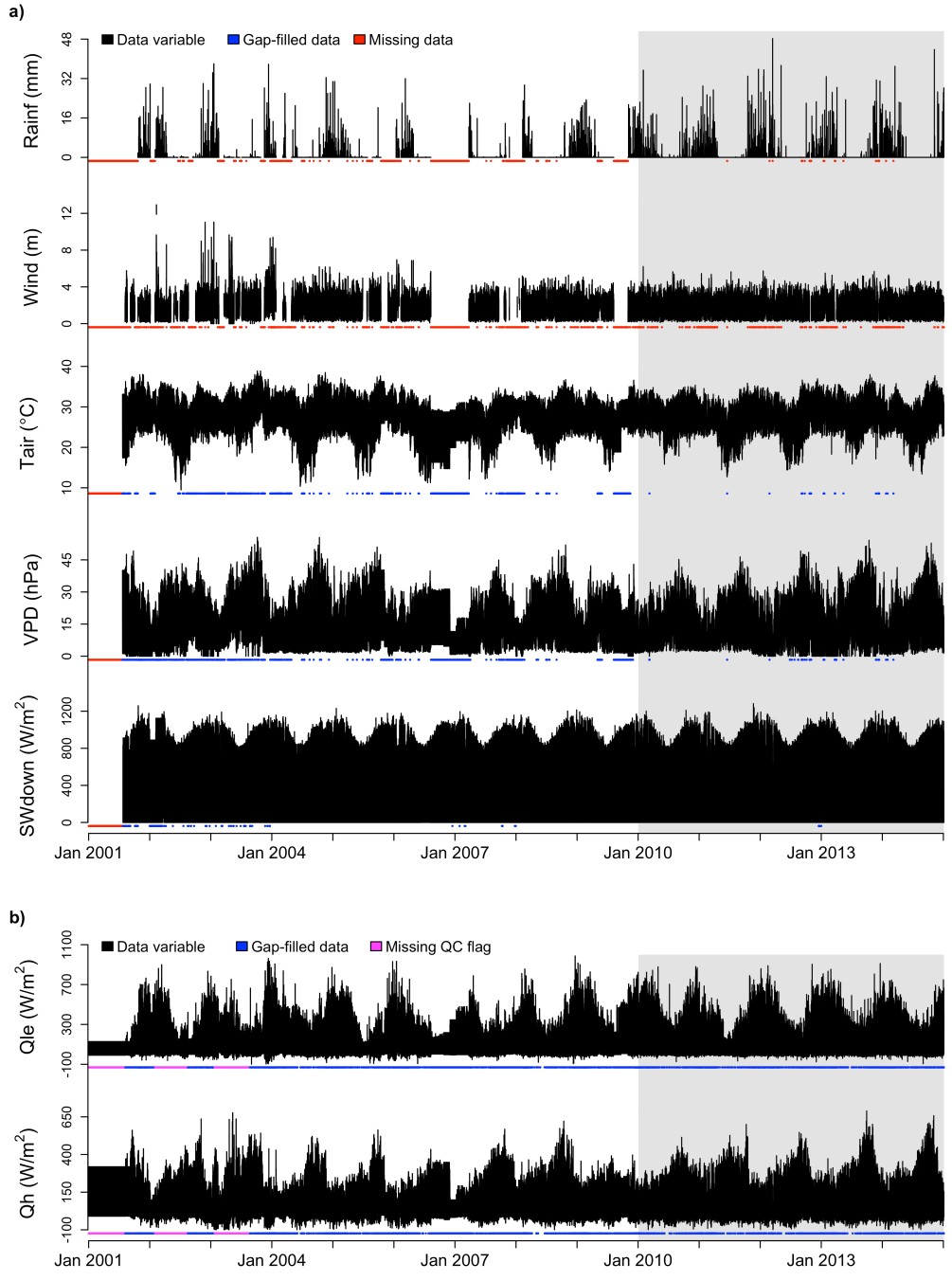

**Figure 3:** Time series of (a) essential meteorological variables and (b) select evaluation variables in Howard Springs. Meteorological variables include precipitation (Rainf), wind speed (Wind), air temperature (Tair), vapour pressure deficit (VPD) and incoming shortwave radiation (SWdown). Latent heat (Qle) and sensible (Qh) are shown as examples of evaluation variables. Gap-filled periods are indicated in blue and missing periods in data variables in red. For evaluation variables, periods with missing quality control (QC) flags are shown in pink.