# Peer review of "FluxnetLSM R package (v1.0): A community tool for processing FLUXNET data for use in land surface modelling"

_Geoscientific Model Development, 2017_

## Referee Comment (RC1) · Anonymous Referee #1 · 28 Apr 2017

The paper submitted by Ukkola et al. "FluxnetLSM R package (v1.0): A community tool for processing FLUXNET data for use in land surface modelling" presents a tool for the transformation and processing of FLUXNET data in order to make them directly available for LSM. The motivation is for sure important for the promotion of use of multiple data streams in LSM validation.

However, the work presented doesn't have any relevant innovative concept or proposal. In fact, despite the import and export functions, change of format to NETCDF, renaming and unit conversions and summary plots (all steps that I don't think limits the use of data in LSM), there are no real innovations.

The gapfilling of the meteorological drivers that is proposed (section 2.4.3) is

an important step where gaps not filled in the timeseries are merged with the ERA-Interim versions, including the creation of a quality indicator. This activity however, looking to the variables description in FLUXNET available at http://fluxnet.fluxdata.org/data/fluxnet2015-dataset/subset-data-product/, is already done in the FLUXNET product (e.g. from the table in the website TA_F = Air temperature, consolidated from TA_F_MDS and TA_ERA, TA_F_QC = Quality flag for TA_F 0 = measured; 1 = good quality gapfill; 2 = downscaled from ERA).

For this reason the paper doesn't have the needed advances, novel concepts, ideas or tools to be considered for publication.

---

## Referee Comment (RC2) · Anonymous Referee #2 · 16 May 2017

Ukkola et al. document an R package to convert FLUXNET data into forcing data for land surface models. This tool might be useful for all land surface models and may lead to more frequent use of FLUXNET data for model evaluation. As the general steps described here are necessary for using FLUXNET data for any land surface model, this can develop into a frequently cited reference.

Reading data files, converting the units and writing them into netcdf is however not a big issue for most scientists. I therefore have some suggestions that could generalize the package more and hopefully lead to a more frequent use of the package.

1) the authors convert the driving data with respect to the units. it might also be useful to provide aggregation to different time steps, not all land surface models use the same

time step in their forcing.

2) The authors only mention that they include the IGBP vegetation classification. Many models however use plant functional types. For the package to be applicable in this respect for most land surface models a conversion to plant functional types would be necessary.

3) It would be interesting to check whether the unit conversion that is applied here is the one required for other models. The authors could gather a list with units for the most widely used land surface models and check whether additional unit conversions are necessary, and if so extend the package accordingly.

Specific comments:

p.2, l. 16-26: I think it would be good to distinguish between the forcing data and the flux measurements used to evaluate the model. The flux measurements do not necessarily need to be gapfilled if the model is compared with these data in high temporal resolution. Then you can simply only use the datapoints that were measured. Of course if you want to evaluate the annual sum the fluxes also need gapfilling.

p.2 l. 35: what are Tier 1 sites?

p.3 l.21: is there any reference for this R package?

p.3 l. 22: "encourages screening of flux tower sites for model applications", what do you mean ? can you be more specific what this screening does?

p.3 l. 26-30: please be more precise: "encourages better documentation", basically this paper is the documentation of the methods, right?

p.7, l. 30: please include all variables that are not gap filled.

p.10, l. 30: did you verify that the format is really directly usable by (many) LSMs? Formats might differ considerably between different models.

p.11, l.1: what are these specific applications?

Reference:

Poulter, B., Ciais, P., Hodson, E., Lischke, H., Maignan, F., Plummer, S., and Zimmermann, N. E.: Plant functional type mapping for earth system models, Geosci. Model Dev., 4, 993-1010, doi:10.5194/gmd-4-993-2011, 2011.

---

## Author Comment (AC2) · 21 Jun 2017

**Response to Reviewer #2**

We would like to thank the reviewer for their helpful comments on our manuscript. In light of the reviewer comments, we propose to include several improvements to the package to provide a more flexible and generalised framework. These include providing:

1) Additional statistical gap-filling methods. The package currently gap-fills forcing variables using ERA-Interim estimates. The new statistical methods (such as linear interpolation or synthesis methods; Abramowitz et al., 2012) will allow the package to be applied to datasets for which ERA-Interim (or similar) estimates are not available and can be used to gap-fill both forcing and evaluation variables.

2) A model-specific look-up table for outputting site vegetation type as a plant functional type in addition to the IGBP vegetation type currently provided by the package.

3) An option to aggregate the data to a different output time step (e.g. daily).

4) The ability to use the package to also process the La Thuile Synthesis Dataset (http://fluxnet.fluxdata.org/data/la-thuile-dataset/). This dataset precedes the FLUXNET2015 data release that the R package is designed for but includes additional sites and may thus be the preferred dataset for some users.

We hope these will further enhance the utility of the package and provide for a wider range of applications. Below we address each of the reviewer comments in more detail.

**General comments:**

Ukkola et al. document an R package to convert FLUXNET data into forcing data for land surface models. This tool might be useful for all land surface models and may lead to more frequent use of FLUXNET data for model evaluation. As the general steps described here are necessary for using FLUXNET data for any land surface model, this can develop into a frequently cited reference. Reading data files, converting the units and writing them into netcdf is however not a big issue for most scientists. I therefore have some suggestions that could generalize the package more and hopefully lead to a more frequent use of the package.

1) the authors convert the driving data with respect to the units. It might also be useful to provide aggregation to different time steps, not all land surface models use the same time step in their forcing.

We thank the reviewer for this suggestion and agree that it would be useful to provide the option to aggregate the forcing data to a different time step. The package currently outputs the data at the same time step as the flux tower data is provided in but we will add an option to customise the time step in the revised package.

We note that the aggregation would require the driving variables (such as precipitation) to be gap-filled in order to perform aggregation. The package currently provides the option to gap-fill these using downscaled ERA-Interim estimates. We propose to also add additional methods for statistical gap-filling (see point 1) above) for applications where ERA-Interim estimates are not available or statistical gap-filling is preferred.

2) The authors only mention that they include the IGBP vegetation classification. Many models however use plant functional types. For the package to be applicable in this respect for most land surface models a conversion to plant functional types would be necessary.

We acknowledge the reviewer's point that a conversion to plant functional types (PFT) may be necessary for other modelling groups. We will therefore add a new functionality to the package to output the vegetation type as a PFT.

As the PFTs used vary between individual models, we do not wish to provide an automated translation between the IGBP vegetation types and PFTs. The IGBP classification also does not distinguish between common model PFT types, such as $C_3$ and $C_4$ grass. Moreover, the choice of vegetation type for each site is model specific and at times a subjective choice. For example, a savanna site, such as Howard Springs, could be modelled as a $C_4$ grass, an evergreen broadleaf tree, or a combination of both PFTs depending on the model configuration and application (De Kauwe et al., 2015; Whitley et al., 2016).

To overcome these challenges, we will provide a model-specific look-up table for PFTs for each site, with the PFT type nominated by the user. This will be integrated with the site metadata file currently provided with the package (P5 L13). This will provide the user with flexibility to set each site's PFT to suit their model and application.

3) It would be interesting to check whether the unit conversion that is applied here is the one required for other models. The authors could gather a list with units for the most widely used land surface models and check whether additional unit conversions are necessary, and if so extend the package accordingly.

Our package uses the ALMA (Assistance for Land surface Modelling Activities; http://www.lmd.jussieu.fr/~polcher/ALMA/) convention. This has been the standard for land surface models since the mid-1990s (Polcher and Shao, 1996) and has been used for a number of model intercomparison projects, such as PILPS (Project for Intercomparison of Land surface Parameterisation Schemes; Lettenmaier, 2003; Polcher and Shao, 1996), and more recently PLUMBER (The Protocol for the Analysis of Land Surface Models (PALS) Land Surface Model Benchmarking Evaluation Project; Best

et al., 2015) and GSWP3 (Global Surface Wetness Project Phase 3; http://hydro.iis.u-tokyo.ac.jp/GSWP3/). The outputs are also CF (Climate and Forecast) compliant. This is the prevailing metadata convention used across the climate science and forecasting community (http://cfconventions.org/). We recognise that some models (e.g. JULES) use a different format, but developing a package that accounts for every eventuality is not feasible. Instead what we have done is develop a generic community tool that could be easily adapted to a specific scenario (e.g. a model not compatible with ALMA or NetCDF). We are of course willing to work with individual groups, helping where possible.

We would also like to note that the output variable names are fully customisable and need not comply with those used in ALMA (as detailed on P6 L5). While only ALMA unit conversions are currently provided, addition of new conversions to the package is trivial. We will provide instructions for this in the package README file (https://github.com/aukkola/FluxnetLSM).

**Specific comments:**

p.2, l. 16-26: I think it would be good to distinguish between the forcing data and the flux measurements used to evaluate the model. The flux measurements do not necessarily need to be gapfilled if the model is compared with these data in high temporal resolution. Then you can simply only use the datapoints that were measured. Of course if you want to evaluate the annual sum the fluxes also need gapfilling.

The flux measurements are not currently gap-filled by the package, as stated on P6 L24. Only meteorological variables are gap-filled using ERA-Interim estimates (if this option is chosen by the user). We propose to add additional statistical gap-filling functions to the package that do not rely on ERA-Interim data to give the user the option to also gap-fill flux variables (see point 1) above). A number of gap-filling options (such as linear interpolation, copy-fill

and synthesis based on other variables) are already provided in the PALS R package and will be integrated with the new package.

p.2 l. 35: what are Tier 1 sites?

The Fluxnet2015 data release has two data tiers with different data usage policies. The Tier 1 sites are those with an open data policy and are thus likely to be those used by the majority of users. We will clarify this in the revised manuscript.

p.3 l.21: is there any reference for this R package?

We will provide the appropriate reference (Abramowitz, 2012) for this package in the revised manuscript, in addition to the github repository already provided.

p.3 l. 22: "encourages screening of flux tower sites for model applications", what do you mean ? can you be more specific what this screening does?

The flux tower data have been gap-filled to various degrees and may have missing data periods. In many circumstances, these are not desirable for modelling applications. Our package provides an automated method for screening gap-filled and missing data. However, this may not detect all data periods and/or sites that are not desired in a particular application. The diagnostic plots generated by the package provide a final quality control step to complement the automated screening to verify that the data are realistic and as expected. For example, this will allow the user to check the magnitude and nature of variability of particular variables. We will clarify this in the revised manuscript.

p.3 l. 26-30: please be more precise: "encourages better documentation", basically this paper is the documentation of the methods, right?

What we intended to say is that the use of the package will allow the data processing methods to be fully reproducible (by including as much metadata as possible in the data files, as well as metadata about the processing used to generate the files) and easily documented in a manuscript. We will clarify this in the revised manuscript.

p.7, l. 30: please include all variables that are not gap filled.

We will name all variables that are not gap-filled in the revised manuscript.

p.10, l. 30: did you verify that the format is really directly usable by (many) LSMs? Formats might differ considerably between different models.

Our package uses the ALMA convention. This has been the standard format for the land surface modelling since the mid-1990s (Polcher and Shao, 1996) and has been used in several previous model intercomparison studies. See our response to reviewer comment #3 above for full details.

p.11, l.1: what are these specific applications?

The applications can range from model evaluation studies to addressing scientific questions using models at the site scales. For example, the user may wish to process the data differently if interested in evaluating models during short-term phenomena (such as heat waves) as opposed to longer seasonal to annual scales. We will clarify this in the revised manuscript.

Reference: Poulter, B., Ciais, P., Hodson, E., Lischke, H., Maignan, F., Plummer, S., and Zimmermann, N. E.: Plant functional type mapping for earth system models, Geosci. Model Dev., 4, 993-1010, doi:10.5194/gmd-4-993-2011, 2011.

References:
Abramowitz, G.: Towards a public, standardized, diagnostic benchmarking system for land surface models, Geosci. Model Dev., 5, 819–827,

doi:10.5194/gmd-5-819-2012, 2012.

Abramowitz, G., Pouyanné, L. and Ajami, H.: On the information content of surface meteorology for downward atmospheric long-wave radiation synthesis, Geophys. Res. Lett., 39, 1–5, doi:10.1029/2011GL050726, 2012.

Best, M. J., Abramowitz, G., Johnson, H. R., Pitman, A. J., Balsamo, G., Boone, A., Cuntz, M., Decharme, B., Dirmeyer, P. A., Dong, J., Ek, M., Guo, Z., Haverd, V., van den Hurk, B. J. ., Nearing, G. S., Pak, B., Peters-Lidard, C., Santanello, J. A., Stevens, L. and Vuichard, N.: The plumbing of land surface models: benchmarking model performance, J. Hydrometeorol., 16, 1425–1442, doi:10.1175/JHM-D-14-0158.1, 2015.

De Kauwe, M. G., Kala, J., Lin, Y.-S., Pitman, A. J., Medlyn, B. E., Duursma, R. A., Abramowitz, G., Wang, Y.-P. and Miralles, D. G.: A test of an optimal stomatal conductance scheme within the CABLE land surface model, Geosci. Model Dev., 8, 431–452, doi:10.5194/gmd-8-431-2015, 2015.
Lettenmaier, D. P.: Preface, Glob. Planet. Change, 38, vii–ix, doi:10.1016/S0921-8181(03)00002-X, 2003.

Polcher, J. and Shao, Y.: A standard format for reporting PILPS experiments, Glob. Planet. Change, 13, 217–223, doi:10.1016/0921-8181(95)00047-X, 1996.

Whitley, R., Beringer, J., Hutley, L., Abramowitz, G., De Kauwe, M. G., Duursma, R., Evans, B., Haverd, V., Li, L., Ryu, Y., Smith, B., Wang, Y.-P., Williams, M. and Yu, Q.: A model inter-comparison study to examine limiting factors in modelling Australian tropical savannas, Biogeosciences, 13, 3245–3265, doi:10.5194/bgd-12-18999-2015, 2016.

---

## Author Response (AR1)

**Response to Reviewer #1**

We would like to thank the reviewer for taking the time to review our manuscript. We provide our point-by-point responses to the reviewer comments below.

**Reviewer comments:**

The paper submitted by Ukkola et al. "FluxnetLSM R package (v1.0): A community tool for processing FLUXNET data for use in land surface modelling" presents a tool for the transformation and processing of FLUXNET data in order to make them directly available for LSM. The motivation is for sure important for the promotion of use of multiple data streams in LSM validation. However, the work presented doesn't have any relevant innovative concept or proposal. In fact, despite the import and export functions, change of format to NETCDF, renaming and unit conversions and summary plots (all steps that I don't think limits the use of data in LSM), there are no real innovations. The gapfilling of the meteorological drivers that is proposed (section 2.4.3) is an important step where gaps not filled in the timeseries are merged with the ERA-Interim versions, including the creation of a quality indicator. This activity however, looking to the variables description in FLUXNET available at http://fluxnet.fluxdata.org/data/fluxnet2015-dataset/subset-data-product/, is already done in the FLUXNET product (e.g. from the table in the website TA_F = Air temperature, consolidated from TA_F_MDS and TA_ERA, TA_F_QC = Quality flag for TA_F 0 = measured; 1 = good quality gapfill; 2 = downscaled from ERA). For this reason the paper doesn't have the needed advances, novel concepts, ideas or tools to be considered for publication.

We agree that the gap-filling of meteorological data is an important step in processing eddy covariance data for use in LSMs. The reviewer is correct that the SUBSET product has been gap-filled using 'good-quality' statistical gap-filling and downscaled ERA-Interim data. However, as the FLUXNET documentation (http://fluxnet.fluxdata.org/data/fluxnet2015-dataset/subset-data-product/) states, the SUBSET product contains minimal data quality and uncertainty information. It is suitable for those looking for an off-the-shelf data product, but does not provide the advanced user with the full resources to produce a dataset fit for purpose. The FULLSET product contains additional quality control information for the statistical gap-filling method used in FLUXNET2015 (Reichstein et al., 2005) that is absent in the SUBSET collection. As such, it provides the user with the full flexibility to use

statistical and ERA-Interim based gap-filling, as facilitated by our R package. An important advantage of our R package is also the possibility to customise the gap-filling methods and add new methods to suit the user's requirements and datasets in a fully citeable and reproducible framework. We will clarify this in the revised manuscript.

In our experience, the discontinuities, varying data quality and incompatible data standards are real challenges for using flux tower data in LSMs. As many of these limitations are often resolved on an *ad hoc* basis, this hinders the reproducibility and transparency of many LSM studies using eddy covariance data, leads to under-utilisation of these data and wasted effort. Our R package aims to overcome these challenges and create a community standard for processing flux tower datasets. The reviewer indicated that they do not consider these as the main limitations hindering the use of flux tower datasets in LSMs. Unfortunately they do not elaborate what these limitations are in their opinion and we are thus unable to address these reviewer concerns in more detail.

**References:**

Reichstein, M., Falge, E., Baldocchi, D., Papale, D., Aubinet, M., Berbigier, P., Bernhofer, C., Buchmann, N., Gilmanov, T., Granier, A., Grünwald, T., Havránková, K., Ilvesniemi, H., Janous, D., Knohl, A., Laurila, T., Lohila, A., Loustau, D., Matteucci, G., Meyers, T., Miglietta, F., Ourcival, J. M., Pumpanen, J., Rambal, S., Rotenberg, E., Sanz, M., Tenhunen, J., Seufert, G., Vaccari, F., Vesala, T., Yakir, D. and Valentini, R.: On the separation of net ecosystem exchange into assimilation and ecosystem respiration: Review and improved algorithm, Glob. Chang. Biol., 11, 1424–1439, doi:10.1111/j.1365-2486.2005.001002.x, 2005.

**Response to Reviewer #2**

We would like to thank the reviewer for their helpful comments on our manuscript. In light of the reviewer comments, we propose to include several improvements to the package to provide a more flexible and generalised framework. These include providing:

1) Additional statistical gap-filling methods. The package currently gap-fills forcing variables using ERA-Interim estimates. The new statistical methods (such as linear interpolation or synthesis methods; Abramowitz et al., 2012) will allow the package to be applied to datasets for which ERA-Interim (or similar) estimates are not available and can be used to gap-fill both forcing and evaluation variables. See section 2.4.3 of the revised manuscript.

2) A model-specific look-up table for outputting site vegetation type as a plant functional type in addition to the IGBP vegetation type currently provided by the package. See section 2.3 of the revised manuscript.

3) An option to aggregate the data to a different output time step (e.g. daily). See section 2.4.5 of the revised manuscript.

4) The ability to use the package to also process the La Thuile Synthesis Dataset (http://fluxnet.fluxdata.org/data/la-thuile-dataset/). This dataset precedes the FLUXNET2015 data release that the R package is designed for but includes additional sites and may thus be the preferred dataset for some users.

We hope these will further enhance the utility of the package and provide for a wider range of applications. Below we address each of the reviewer comments in more detail.

**General comments:**

Ukkola et al. document an R package to convert FLUXNET data into forcing data for land surface models. This tool might be useful for all land surface models and may lead to more frequent use of FLUXNET data for model evaluation. As the general steps described here are necessary for using FLUXNET data for any land surface model, this can develop into a frequently cited reference. Reading data files, converting the units and writing them into netcdf is however not a big issue for most scientists. I therefore have some suggestions that could generalize the package more and hopefully lead to a more frequent use of the package.

1) the authors convert the driving data with respect to the units. It might also be useful to provide aggregation to different time steps, not all land surface models use the same time step in their forcing.

We thank the reviewer for this suggestion and agree that it would be useful to provide the option to aggregate the forcing data to a different time step. The package currently outputs the data at the same time step as the flux tower data is provided in but we have added an option to customise the time step in the revised package up to a daily resolution:

*"By default, the package outputs the data in its original time resolution. However, a longer time step may be desired for some model applications. The package allows the aggregation of the data to up to a daily resolution. The aggregated time step size (in hours) is set by the argument aggregate and can be any number between the original resolution (usually 30 minutes) and 24 hours (daily), as long as it is divisible by 24 to allow a regular number of time steps to be aggregated. If any of the time steps being aggregated are missing, the new coarser time step will also be set to missing. The QC flags (if outputted) are assigned a fraction between 0-1, indicating the percentage of time steps used for aggregation that were observed."*

2) The authors only mention that they include the IGBP vegetation classification. Many models however use plant functional types. For the package to be applicable in this respect for most land surface models a conversion to plant functional types would be necessary.

We acknowledge the reviewer's point that a conversion to plant functional types (PFT) may be necessary for other modelling groups. We have therefore added a new functionality to the package to output the vegetation type as a PFT.

As the PFTs used vary between individual models, we do not wish to provide an automated translation between the IGBP vegetation types and PFTs. The IGBP classification also does not distinguish between common model PFT types, such as $C_3$ and $C_4$ grass. Moreover, the choice of vegetation type for each site is model specific and at times a subjective choice. For example, a savanna site, such as Howard Springs, could be modelled as a $C_4$ grass, an evergreen broadleaf tree, or a combination of both PFTs depending on the model configuration and application (De Kauwe et al., 2015; Whitley et al., 2016).

To overcome these challenges, we will provide a model-specific look-up table for PFTs for each site, with the PFT type nominated by the user. This will be integrated

with the site metadata file currently provided with the package (P5 L13). This will provide the user with flexibility to set each site's PFT to suit their model and application:

*"This processing step connects key site metadata directly to each model forcing files. It can be extended to include additional metadata, such as site soil or vegetation properties, with minimal code modifications. For example, LSMs generally use plant functional types (PFT) instead of the International Geosphere-Biosphere Programme (IGBP; http://www.igbp.net/) vegetation types automatically retrieved by the package (Poulter et al., 2011). An example is provided for writing the PFT type for the CABLE LSM (Wang et al., 2011) and can be invoked by setting the model argument to the desired model name. Full instructions for adding model-specific parameters are provided in the package README file."*

3) It would be interesting to check whether the unit conversion that is applied here is the one required for other models. The authors could gather a list with units for the most widely used land surface models and check whether additional unit conversions are necessary, and if so extend the package accordingly.

Our package uses the ALMA (Assistance for Land surface Modelling Activities; http://www.lmd.jussieu.fr/~polcher/ALMA/) convention. This has been the standard for land surface models since the mid-1990s (Polcher and Shao, 1996) and has been used for a number of model intercomparison projects, such as PILPS (Project for Intercomparison of Land surface Parameterisation Schemes; Lettenmaier, 2003; Polcher and Shao, 1996), and more recently PLUMBER (The Protocol for the Analysis of Land Surface Models (PALS) Land Surface Model Benchmarking Evaluation Project; Best et al., 2015) and GSWP3 (Global Surface Wetness Project Phase 3; http://hydro.iis.u-tokyo.ac.jp/GSWP3/). The outputs are also CF (Climate and Forecast) compliant. This is the prevailing metadata convention used across the climate science and forecasting community (http://cfconventions.org/). We recognise that some models (e.g. JULES) use a different format, but developing a package that accounts for every eventuality is not feasible. Instead what we have done is develop a generic community tool that could be easily adapted to a specific scenario (e.g. a model not compatible with ALMA or NetCDF). We are of course willing to work with individual groups, helping where possible.

We would also like to note that the output variable names are fully customisable and need not comply with those used in ALMA (as detailed on P6 L5). While only ALMA unit conversions are currently provided, addition of new conversions to the package is trivial. We will provide instructions for this in the package README file (https://github.com/aukkola/FluxnetLSM).

**Specific comments:**

p.2, l. 16-26: I think it would be good to distinguish between the forcing data and the flux measurements used to evaluate the model. The flux measurements do not necessarily need to be gapfilled if the model is compared with these data in high temporal resolution. Then you can simply only use the datapoints that were measured. Of course if you want to evaluate the annual sum the fluxes also need gapfilling.

The flux measurements are not currently gap-filled by the package. Only meteorological variables are gap-filled using ERA-Interim estimates (if this option is chosen by the user). We have added additional statistical gap-filling functions to the package that do not rely on ERA-Interim data to give the user the option to also gap-fill flux variables (see point 1) above). A number of gap-filling options (such as linear interpolation, copy-fill and synthesis based on other variables) are already provided in the PALS R package and have been integrated with the new package (see section 2.4.3 of the revised manuscript).

p.2 l. 35: what are Tier 1 sites?

The Fluxnet2015 data release has two data tiers with different data usage policies. The Tier 1 sites are those with an open data policy and are thus likely to be those used by the majority of users. We have clarified this in the revised manuscript (P2 L39).

p.3 l.21: is there any reference for this R package?

We have removed dependency on the PALS R package after identifying difficulties installing this package. Required PALS functions are now replicated in the FluxnetLSM package.

p.3 l. 22: "encourages screening of flux tower sites for model applications", what do you mean ? can you be more specific what this screening does?

The flux tower data have been gap-filled to various degrees and may have missing data periods. In many circumstances, these are not desirable for modelling applications. Our package provides an automated method for screening gap-filled and missing data. However, this may not detect all data periods and/or sites that are not desired in a particular application. The diagnostic plots generated by the package provide a final quality control step to complement the automated screening to verify that the data are realistic and as expected. For example, this will allow the user to check the magnitude and nature of variability of particular variables. We have clarified this in the revised manuscript:

*"This facilitates the detection of data periods with unusual variability or variables exhibiting unusual magnitudes"*

p.3 l. 26-30: please be more precise: "encourages better documentation", basically this paper is the documentation of the methods, right?

What we intended to say is that the use of the package will allow the data processing methods to be fully reproducible (by including as much metadata as possible in the data files, as well as metadata about the processing used to generate the files) and easily documented in a manuscript. We have clarified this in the revised manuscript.

*"The package offers a useful tool for post-processing eddy covariance datasets for modelling applications and simplifies rigorous documentation of data processing methods in LSM studies to enhance their reproducibility. Specifically, future studies using these data would be able to explicitly demonstrate how the data were used, gap-filled, quality controlled and so on, and this could be reproduced by other users."*

p.7, l. 30: please include all variables that are not gap filled.

We have named all variables that are not gap-filled in the revised manuscript.

p.10, l. 30: did you verify that the format is really directly usable by (many) LSMs? Formats might differ considerably between different models.

Our package uses the ALMA convention. This has been the standard format for the land surface modelling since the mid-1990s (Polcher and Shao, 1996) and has been used in several previous model intercomparison studies. See our response to reviewer comment #3 above for full details.

p.11, l.1: what are these specific applications?

The applications can range from model evaluation studies to addressing scientific questions using models at the site scales. For example, the user may wish to process the data differently if interested in evaluating models during short-term phenomena (such as heat waves) as opposed to longer seasonal to annual scales. We have clarified this in the revised manuscript.

*"Simultaneously, it provides optional settings for an advanced user to produce flux tower datasets suited for specific applications. For example, the user may wish to process the data differently if interested in evaluating models during short-term phenomena (such as heat waves) compared to longer seasonal to annual scales."*

Reference: Poulter, B., Ciais, P., Hodson, E., Lischke, H., Maignan, F., Plummer, S., and Zimmermann, N. E.: Plant functional type mapping for earth system models, Geosci. Model Dev., 4, 993-1010, doi:10.5194/gmd-4-993-2011, 2011.

[revised manuscript text omitted]